# Role of the Ca²⁺ channel α₂δ-1 auxiliary subunit in proliferation and migration of human glioblastoma cells

**Miriam Fernández-Gallardo**[1], **Alejandra Corzo-Lopez**[2], **David Muñoz-Herrera**[2], **Margarita Leyva-Leyva**[3], **Ricardo González-Ramírez**[3], **Alejandro Sandoval**[4], **Rodolfo Delgado-Lezama**[1], **Eduardo Monjaraz**[5], **Ricardo Felix**[2]*

**1** Department of Physiology, Biophysics and Neuroscience, Center for Research and Advanced Studies of the National Polytechnic Institute (Cinvestav-IPN), Mexico City, Mexico, **2** Department of Cell Biology, Cinvestav-IPN, Mexico City, Mexico, **3** Department of Molecular Biology and Histocompatibility, "Dr. Manuel Gea González" General Hospital, Mexico City, Mexico, **4** School of Medicine FES Iztacala, National Autonomous University of Mexico (UNAM), Tlalnepantla, Mexico, **5** Institute of Physiology, Meritorious Autonomous University of Puebla (BUAP), Puebla, Mexico

* rfelix@cinvestav.mx

**Data Availability Statement:** All relevant data are within the manuscript and its Supporting Information files.

## Abstract

The overexpression of α₂δ-1 is related to the development and degree of malignancy of diverse types of cancer. This protein is an auxiliary subunit of voltage-gated Ca²⁺ (Ca$_V$) channels, whose expression favors the trafficking of the main pore-forming subunit of the channel complex (α₁) to the plasma membrane, thereby generating an increase in Ca²⁺ entry. Interestingly, TLR-4, a protein belonging to the family of toll-like receptors that participate in the inflammatory response and the transcription factor Sp1, have been linked to the progression of glioblastoma multiforme (GBM). Therefore, this report aimed to evaluate the role of the α₂δ-1 subunit in the progression of GBM and investigate whether Sp1 regulates its expression after the activation of TLR-4. To this end, the expression of α₂δ-1, TLR-4, and Sp1 was assessed in the U87 human glioblastoma cell line, and proliferation and migration assays were conducted using different agonists and antagonists. The actions of α₂δ-1 were also investigated using overexpression and knockdown strategies. Initial luciferase assays and Western blot analyses showed that the activation of TLR-4 favors the transcription and expression of α₂δ-1, which promoted the proliferation and migration of the U87 cells. Consistent with this, overexpression of α₂δ-1, Sp1, and TLR-4 increased cell proliferation and migration, while their knockdown with specific siRNAs abrogated these actions. Our data also suggest that TLR-4-mediated regulation of α₂δ-1 expression occurs through the NF-kB signaling pathway. Together, these findings strongly suggest that the activation of TLR-4 increases the expression of α₂δ-1 in U87 cells, favoring their proliferative and migratory potential, which might eventually provide a theoretical basis to examine novel biomarkers and molecular targets for the diagnosis and treatment of GBM.

**Funding:** The authors received no specific funding for this work.

**Competing interests:** The authors have declared that no competing interests exist.

## Introduction

Glioblastoma multiforme (GBM) is the most aggressive primary brain tumor, accounting for >70% of all brain tumors in adults over 50 years of age. Despite the various treatments for GBM (surgery, radio, and chemotherapy), it remains a disease with a poor prognosis for survival [1, 2]. Indeed, despite the development in research on different molecular aspects of GBM, the tumor continues to have one of the highest mortality rates; consequently, there is a growing interest in learning more regarding its origin and development.

On the other hand, $Ca^{2+}$ is a ubiquitous divalent cation and a second messenger that participates in multiple physiological processes regulated by transient changes in its intracellular concentration. Therefore, intracellular calcium concentrations are determined by an elaborate system of transporters, channels, membrane exchangers, binding/buffer proteins, and ATPases that finely regulate the flow of this cation in and out of cells and between different cellular organelles [3, 4]. When an alteration occurs in these systems, changes are caused in the dynamics of $Ca^{2+}$ flows, affecting the functioning of various cellular events such as proliferation, gene expression, protein phosphorylation/dephosphorylation, and cell death [4–6].

In this context, voltage-gated $Ca^{2+}$ ($Ca_V$) channels have been shown to play a fundamental role in tumor biology. Specifically, these channels are known to exhibit altered functional expression or aberrant localization associated with the promotion of tumor growth and cell migration, as well as resistance to treatment [4–8]. $Ca_V$ channels are transmembrane oligomeric complexes conformed by an ion-conducting $\alpha_1$ subunit and auxiliary $\alpha_2\delta$ and β subunits that are activated by cell membrane depolarization. $Ca_V$ channels conduct $Ca^{2+}$ into cells, where they initiate a wide variety of physiological responses, ranging from hormone and neurotransmitter release and muscle contraction to gene transcription, among many others [9]. In addition, these channels often associate with other proteins creating micro or nanodomains of $Ca^{2+}$ signaling [10–12].

It has been previously reported also that the overexpression of the protein $\alpha_2\delta$-1 is related to the development and degree of malignancy of numerous types of cancer, such as larynx, lung, ovary, liver, stomach, and breast [13–20]. In addition, it is known that the promoter region of the *CACNA2D1* gene, which encodes for $\alpha_2\delta$-1, has two functional Sp1 binding sites [21, 22]. Interestingly, various reports have shown the over-expression of Sp1 in diverse types of cancer, relating its expression level to tumor stage and unfavorable prognosis for patient survival [23–25]. Likewise, the silencing or inhibition of Sp1 reduces tumor formation, growth, and metastasis [26]. Interestingly, it has been reported that the expression of Sp1 in tumor cells induces the expression of diverse growth factors related to tumor growth and metastasis [27]. Hence, Küper et al. (2012), using renal medullary collecting duct cells, found that activation of the toll-like receptor TLR-4 promoted the activity of the transcription factor NF-κB, which participates in the regulation of Sp1 expression [28].

Based on these findings, in the present report, we evaluated the contribution of $\alpha_2\delta$-1 to the proliferation and migration processes of U87 human glioblastoma cells and its possible regulation through the activation of TLR-4 and Sp1 signaling pathway. Our results support the idea that the activation of TLR-4 triggers the nuclear translocation of the transcription factor NF-κB, with the consequent increase in Sp1 expression, which regulates the presentation of $\alpha_2\delta$-1.

## Materials and methods

### Cell culture

The human glioblastoma cell line U87MG and human neuroblastoma SH-SY5Y were purchased from the American Type Culture Collection (ATCC). Cells were grown in monolayer

in Dulbecco's Modified Eagle Culture Medium (DMEM, Thermo Fisher Scientific) supplemented with 10% fetal bovine serum and 100 μg/mL streptomycin and 100 μl/mL penicillin. Cells were kept in a 37˚C incubator with 5% $CO_2$ and passaged once a week.

## Protein extraction and Western blot analysis

U87 human glioblastoma cells were seeded in 60 mm Petri dishes until approximately 90% confluency and washed with phosphate-buffered saline (PBS). Subsequently, 250 μL of radio-immunoprecipitation assay (RIPA) lysis buffer containing (in mM): 100 Tris-HCl (pH 8.0), 150 NaCl, 1 EDTA (pH 8.0), 1% Triton X-100, 0.5% sodium deoxycholate, 0.1% SDS, 1-phenylmethylsulfonyl fluoride (PMSF), and Complete 1 (Roche). Samples were subsequently incubated on ice for 30 min, and the cell lysate was subjected to centrifugation at 12,000 X g for 2 min to collect the supernatant. Thirty μg of protein were suspended in Laemmli 1 buffer solution (SDS 1.6%, 2-mercaptoethanol 0.1 M, glycerol 5%, 4Tris-Cl/SDS 0.083 M (pH 6.8) and bromophenol blue at 0.002%) and used for electrophoresis. Samples were heated at 95˚C for 5 min, and proteins were electrophoresed on an 8% polyacrylamide gel in electrophoresis buffer (0.025 M Tris base, 0.192 M glycine, 0.1% SDS). After separation, proteins were transferred to a 0.45 μm nitrocellulose membrane and stained with Ponceau red. After washing the membrane with Tris-buffered saline (TBS)-Tween 20 (TBST) and blocking with 5% nonfat milk in TBST for 2 h at room temperature, membranes were incubated overnight at 4˚C with the antibodies anti-$\alpha_2\delta$-1 (1:1000), anti-Sp1 (1:1000), anti-TLR-4 (1:500), and anti-β-actin (1:10,000) (Santa Cruz Biotechnology). Secondary antibodies used were anti-mouse (1:10,000) or anti-rabbit (1:10,000), and protein immunodetection was performed using a commercial chemiluminescence kit (Millipore) and imaged with the Odyssey Fc imaging system (LI-COR).

## Wound healing assays

U87 cells were plated in 35 mm tissue culture dishes at a density of $1 \times 10^6$ to achieve ~90% confluency. The cell monolayer was then wounded with a two hundred μL pipette tip. Cultures were washed with PBS and DMEM medium with 10% FBS three times. Subsequently, the dishes with the cell cultures were placed in a Nikon Eclipse TE300 inverted microscope to obtain the images. Using a Nikon Plan Apo 4X/0.2 objective and digital camera, various random fields were selected for imaging. Images were captured at appropriate times to assess wound closure. The relative distance of wound closure estimated the efficacy of lipopolysaccharide (LPS) stimulation on the migratory capacity of cells.

## Luciferase assays

Twenty-four h after seeding, cells at $\sim$80% confluency were subjected to a transient transfection using Lipofectamine 2000 reagent (Invitrogen) according to the manufacturer's protocol. Cells were co-transfected with 2 μg of the $\alpha_2\delta$-1 promoter construct and 0.4 μg of the pRSV-βGal reporter vector, encoding the β-galactosidase gene under the control of the Rous sarcoma virus promoter, to monitor the efficiency of the transfection. Luciferase activity was quantified in triplicate 48 h after transfection, using a commercial kit (Promega) and a luminometer (Turner BioSystems). To correct for differences in transfection efficiency, luciferase activity was normalized to β-galactosidase values in each sample.

## Cell proliferation assays

Cell proliferation assays were performed using the Countess II FL Automated Cell Counter (Thermo Fisher Scientific). To this end, an aliquot of the U87 cell culture was first obtained.

Previously, the cells were seeded at a density of 1 x $10^6$/mL in 35 mm Petri dishes and kept in culture until reaching a confluence of ~60–70%. Subsequently, the different treatments with the agonists and antagonists or the silencing and overexpression maneuvers were initiated. Forty-eight after treatment, the samples were analyzed. The cell culture was first trypsinized, and 50 μL of the cell suspension was taken. Next, 50 μL of trypan blue at 0.4% was added to the cell suspension. The samples were then homogenized, and subsequently, 10 μL were loaded into the cell counting chamber to determine the percentages of viable and dead cells.

## Cell migration assays

Cell migration assays were performed using 24-well transwell chambers containing 8 μm poly-carbonate nuclepore filters. Filters were initially washed with serum-free DMEM and plated into 24-well chambers. The lower chambers contained DMEM supplemented with 3% FBS. A cell count was initially performed in each experimental condition for the upper chambers to place the same number of cells (3 x $10^4$) in 300 μL of FBS-free DMEM medium. The chambers were then incubated at 37˚C with 5% $CO_2$. After 12 h, cells that did not migrate were removed from the chambers, while cells that migrated through the membranes were fixed with a 1:1 methanol-acetone solution and stained with 0.5% crystal violet. The number of migrating cells was estimated by counting at least ten randomly selected visual fields.

## Small interfering RNA knockdown

U87 cells were seeded at ~80% confluency in 35 mm tissue culture dishes and transfected with a 150 pmol of pre-designed synthetic small interfering RNAs (siRNAs) directed against α₂δ-1, Sp1, TLR- 4 (sc-89621, sc-29487, and sc-40260, respectively), or with a siRNA scramble used as a negative control (Santa Cruz Biotechnology), using the Lipofectamine 2000 transfection reagent (Invitrogen) according to the instructions provided by the manufacturer. Twenty-four hours after transfection, cells were harvested to analyze siRNAs' effect on the corresponding proteins' levels by Western blot. Similarly, Western blot analysis using α₂δ-1-specific antibodies was performed to determine whether Sp1 knockdown affected the α₂δ-1 protein expression.

## Results

### Expression of α₂δ-1, TLR-4–1, and Sp1 in the U87 human glioblastoma cell line

To study the role of the Ca$_V$ channel α₂δ-1 subunit in the proliferative and migratory capacity of U87 cells, and whether the regulation of its expression was dependent on Sp1 after activation of TLR-4, we initially investigated the expression of these proteins in the cell line by Western blot, using antibodies available from commercial sources (see Material and Methods Section) and the human neuroblastoma SHSY-5Y cells as a positive control. Previous reports have documented the expression of α₂δ-1, TLR-4, and Sp1 in the SHSY-5Y cell line [21, 29]. The results of the Western blot analysis showed the presence of strong bands at the predicted molecular weight for α₂δ-1, TLR-4, and Sp1 (~150, ~95, and ~90 kDa, respectively) in the lysates of the two cell lines studied (S1 Fig), corroborating the expression of these three proteins in U87 human glioblastoma cells.

### TLR-4 activation promotes the expression of α₂δ-1

We next evaluated whether the activation of TLR-4 could regulate the transcription and expression of α₂δ-1. To this end, luciferase assays were conducted to assess the activity of the α₂δ-1 promoter in the presence and absence of the TLR-4 activator, LPS. The U87 cells were

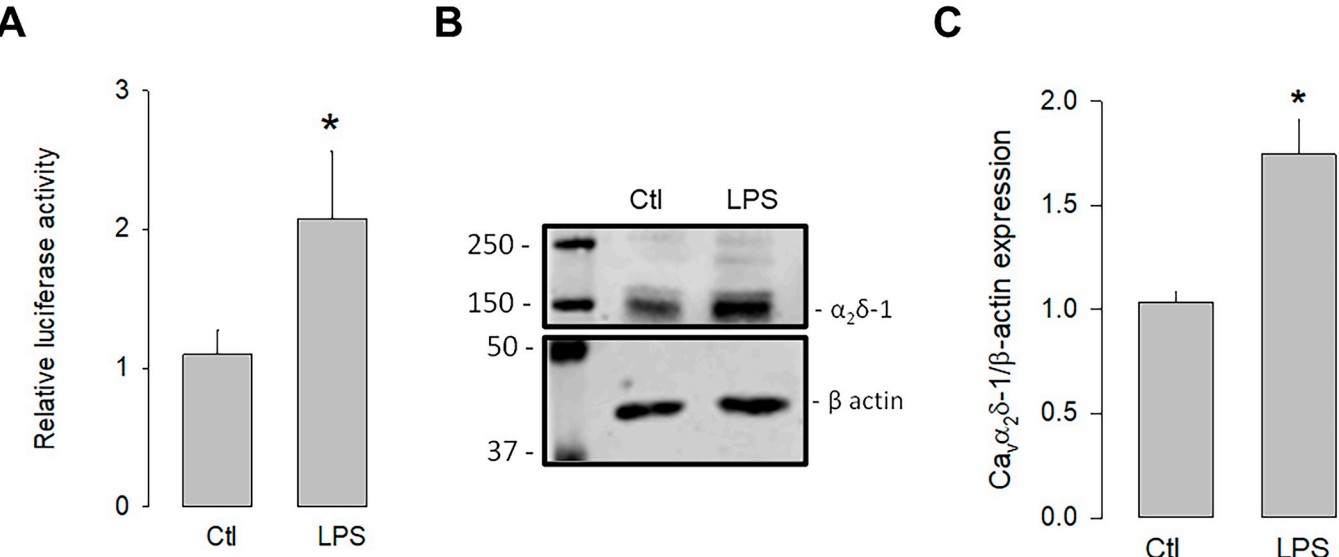

**Fig 1. Upregulation of α$_2$δ-1 by LPS.** U87 cells were co-transfected with a vector containing the α$_2$δ-1 subunit promoter and the *Renilla* luciferase coding sequence (as a reporter gene), and the pRSV-βGal vector, which included the RSV promoter and the β-gal coding sequence to correct for differences related to transfection efficiency. Cells were incubated for 48 h in the absence and presence of the TLR-4 activator, LPS. B) Western blot assay performed on U87 cell lysates in the absence and presence of LPS. The image shows a representative assay of three performed separately. C) Densitometric analysis of protein α$_2$δ-1 normalized with respect to the expression of β actin and depicted as a fold change from the control. Asterisks denote statistically significant differences (P < 0.05) with respect to the control (Ctl).

transfected with an expression vector containing the α$_2$δ-1 promoter and luciferase as a reporter gene, and the cells were treated with LPS for 48 h to evaluate the promoter activity. The results showed that the activation of TLR-4 with LPS (5 µg/mL) significantly increased luciferase activity compared to the control (Fig 1A). In an additional series of experiments, the effect of the LPS treatment on the expression of the α$_2$δ-1 protein was evaluated. Fig 1B shows a representative image of a semiquantitative Western blot experiment, where a higher density band is observed in the lysates from U87 cells treated with LPS compared to the loading control (β-actin). Last, Fig 1C shows that, on average, the level of α$_2$δ-1 protein expression was ∼1.8-fold higher in LPS-treated U87 human glioblastoma cells compared to untreated controls. These data suggest that the activation of TLR-4 by LPS may trigger signaling pathways that positively regulate the expression of α$_2$δ-1.

### TLR-4 activation favors U84 cells proliferation and migration

To study the effects of the TLR-4 activation with LPS on the proliferative capacity, U87 cells were counted automatically after 48 h of incubation in the absence and presence of the lipopolysaccharide. Fig 2A shows that the receptor activation caused a significant increase (>3-fold) in the proliferative capacity of U87 cells compared to the controls. Likewise, to corroborate that the effect produced by the treatment with LPS was the result of the activation of TLR-4, a specific antagonist of the receptor called C34 was used, and the results showed that in the presence of C34, the effect of LPS on cell proliferation were significantly prevented (Fig 2A).

The effects of LPS on the migration of the U87 human glioblastoma cells were then evaluated semiquantitatively using wound healing and migration assays in transwell chambers. After LPS treatment (48 h), the cell monolayer was wounded, and 12 h later, the culture dishes were fixed and stained for analysis. The results show that in the presence of 5 µg/mL of LPS, the wound area was significantly smaller than the controls. This is even more evident at a higher concentration of the lipopolysaccharide (10 µg/mL; Fig 2B).

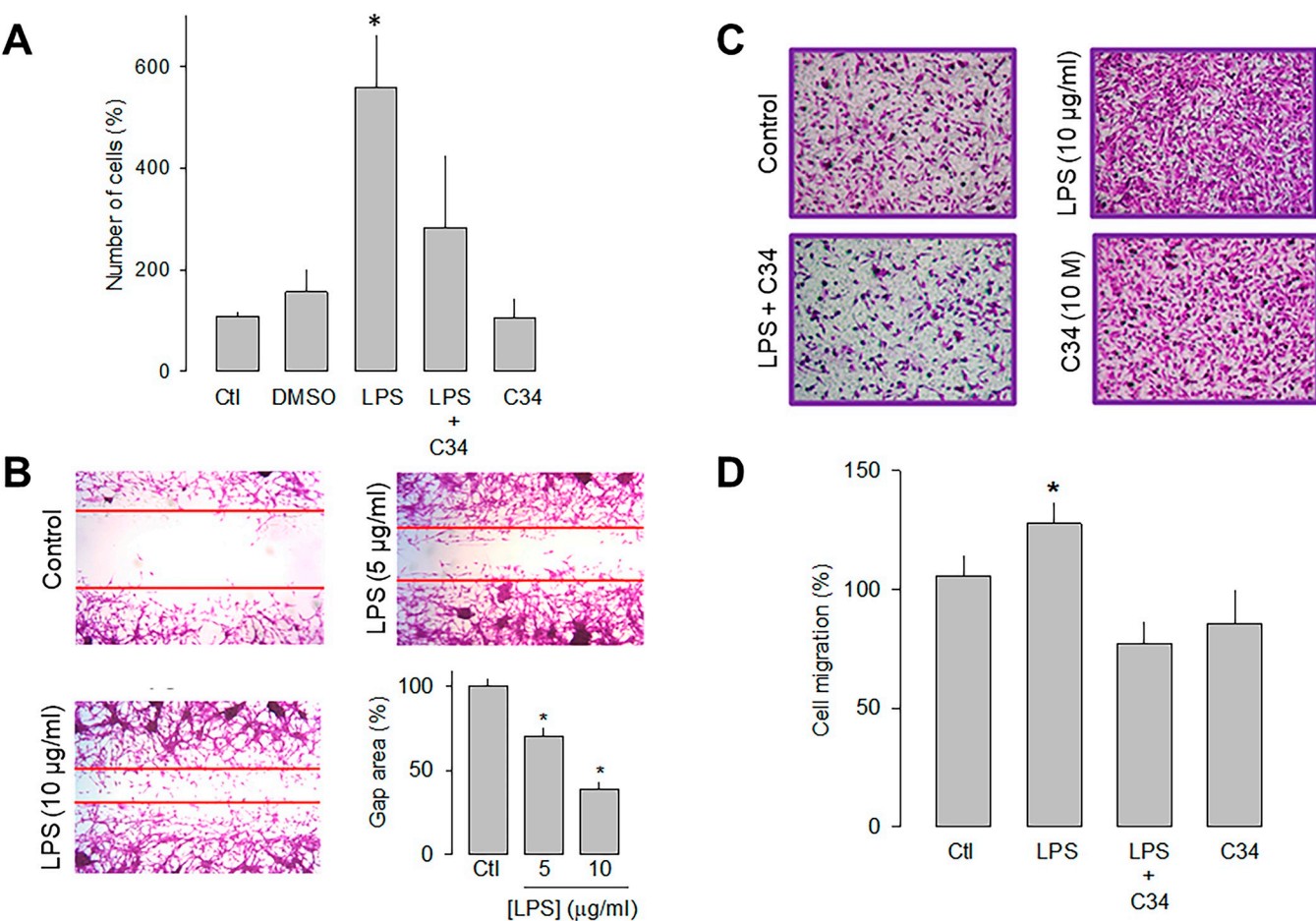

**Fig 2. Effect of LPS on the proliferation and migration of U87 cells.** A) Comparison of direct and automated cell counting of U87 cells maintained 48 h in culture in the presence and absence of the TLR-4 activator, LPS, as indicated. Cell counts were performed in parallel after treatment with C34, a specific inhibitor of TLR-4, and the vehicle (DMSO) in which it was dissolved. Values are expressed as a percent of control, and each bar represents the mean ± SE of triplicate determinations in 3 separate experiments. B) Cell wound healing assays showing the ability of U87 cells to migrate in culture in the absence and presence of LPS, as indicated. The lower right panel shows a graph of the quantification of the wound area. C) Transwell migration assays of U87 cells in the presence of the activator (LPS) and the inhibitor (C34) of TLR-4, as listed. Typical images from three separate experiments are shown. D, Comparative analysis of transwell migration assay results of U78 cells as in C. Results represent the mean ± SEM of 3 independent experiments. *P < 0.05 compared to untreated cells.

On the other hand, the effect of LPS on cell migration was studied, using the antagonist C34 to verify that the activation of TLR-4 was indeed involved in this effect. Representative images of four independent experiments under the different experimental conditions (control, LPS, LPS+C34, and C34 alone) are shown in Fig 2C, and the summary of the results is shown in Fig 2D. This data confirmed that U87 glioblastoma cell migration is favored in the presence of LPS. Interestingly, as occurred in the wound healing assay, the presence of the TLR-4 antagonist prevents the stimulatory effect of LPS. Together, these data show that the activation of TLR-4 may promote the migration process of U87 cells.

## The overexpression of α2δ-1 increases cell proliferation and migration

Once the effect of LPS on the proliferation of U87 cells was evaluated, we next sought to define the effect of the α2δ-1 Cav channel auxiliary subunit overexpression on cell proliferation. Initial experiments showed that 1 μg/μL was determined to be the optimal concentration for

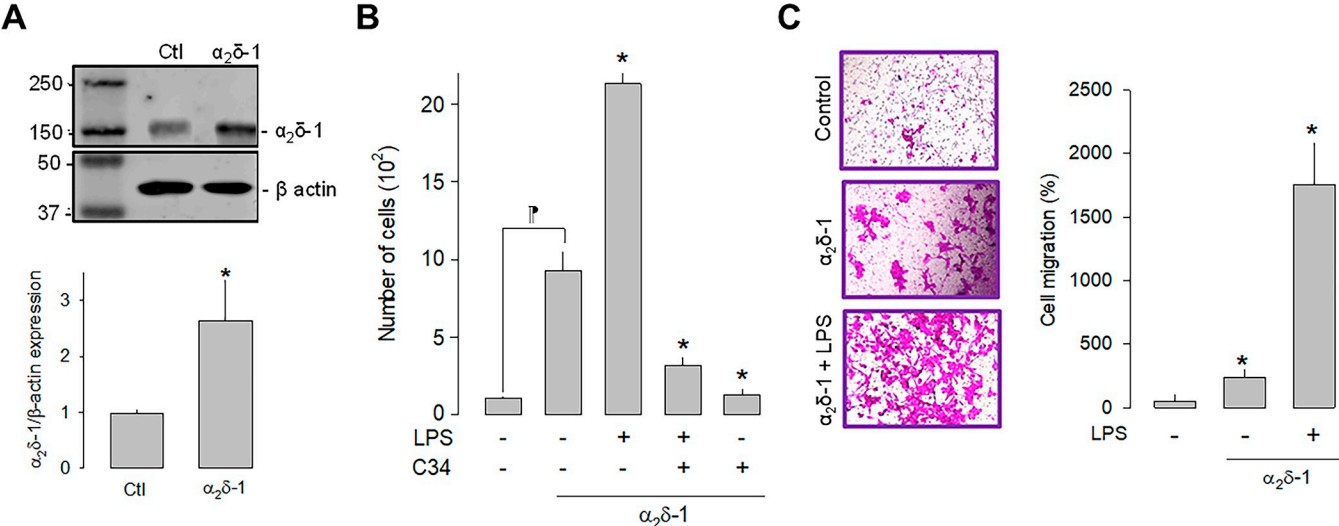

**Fig 3. Effect of $\alpha_2\delta$-1 overexpression on the proliferation and migration of U87 cells.** A) Western blot analysis of U87 cell lysates in control and $\alpha_2\delta$-1 transiently transfected cells. The image in the upper panel shows a representative assay of three performed separately. The lower panel shows the densitometric analysis of the protein $\alpha_2\delta$-1 normalized with respect to the expression of β actin and depicted as a fold change from the control. Asterisks denote a statistically significant difference ($P < 0.05$) with respect to the control (Ctl). B) Comparison of direct and automated cell counting of U87 cells maintained in culture after $\alpha_2\delta$-1 transfection in the presence and absence of the agonist and the antagonist of TLR-4, LPS, and C34, respectively. Values are expressed as the number of cells in each experimental condition, and each bar represents the mean ± SEM of triplicate determinations in 3 separate experiments. *$P < 0.05$ *versus* $\alpha_2\delta$-1 overexpressing cells (second bar from the left). ¶$P < 0.05$ *versus* untransfected cells. C) Transwell migration assays of U87 cells in the presence of the TLR-4 activator (LPS). The left panel shows representative images from three separate experiments. The right panel shows the comparative analysis of the mean ± SEM of 3 transwell independent experiments. *$P < 0.01$ compared to untreated cells.

plasmid transfection and was used in subsequent experiments. The upper panel in Fig 3A shows a representative Western blot image of the overexpression of $\alpha_2\delta$-1, while the bottom panel summarizes the results of three separate experiments. These results corroborate the expression of the $\alpha_2\delta$-1 subunit in the U87 cells and show a significant increase in the protein after transfection with the plasmid encoding the $Ca_V$ channel auxiliary subunit.

We next investigated whether the activation or the inhibition of TLR-4, by LPS or C-34, respectively, were affected by $\alpha_2\delta$-1 overexpression. This analysis showed that the overexpression of the $Ca_V$ channel subunit *per se* significantly increases the number of cells and that this effect is more evident in the presence of LPS. As expected, the effect of LPS is prevented when C34 is present (Fig 3B). These findings suggest that $\alpha_2\delta$-1 favors cell proliferation via activation of TLR-4 in U87 glioblastoma cells. Similarly, cell migration was favored after overexpression of $\alpha_2\delta$-1. The results of the cell migration assays in transwell chambers under control conditions and after overexpression (in the presence and absence of LPS) showed a significantly higher number of cells when $\alpha_2\delta$-1 was overexpressed (Fig 3C).

## Inhibition of $\alpha_2\delta$-1 prevents the effect of LPS on cell migration

Previous studies have shown that the $\alpha_2\delta$-1 subunit increases the cell membrane expression of native and recombinant $Ca_V$ channels. Likewise, it is also known that this protein makes the channels sensitive to the inhibitory effects of a group of antiepileptic/analgesic drugs called gabapentinoids, which includes gabapentin, pregabalin, and AdGABA [30, 31]. Given that the data in the preceding section implicate a role of $\alpha_2\delta$-1 in cell migration, we next sought to explore whether gabapentin was also able to affect the migration of U87 cells. The results of this analysis showed that chronic exposure (48 h) to gabapentin (25 µM) prevented the

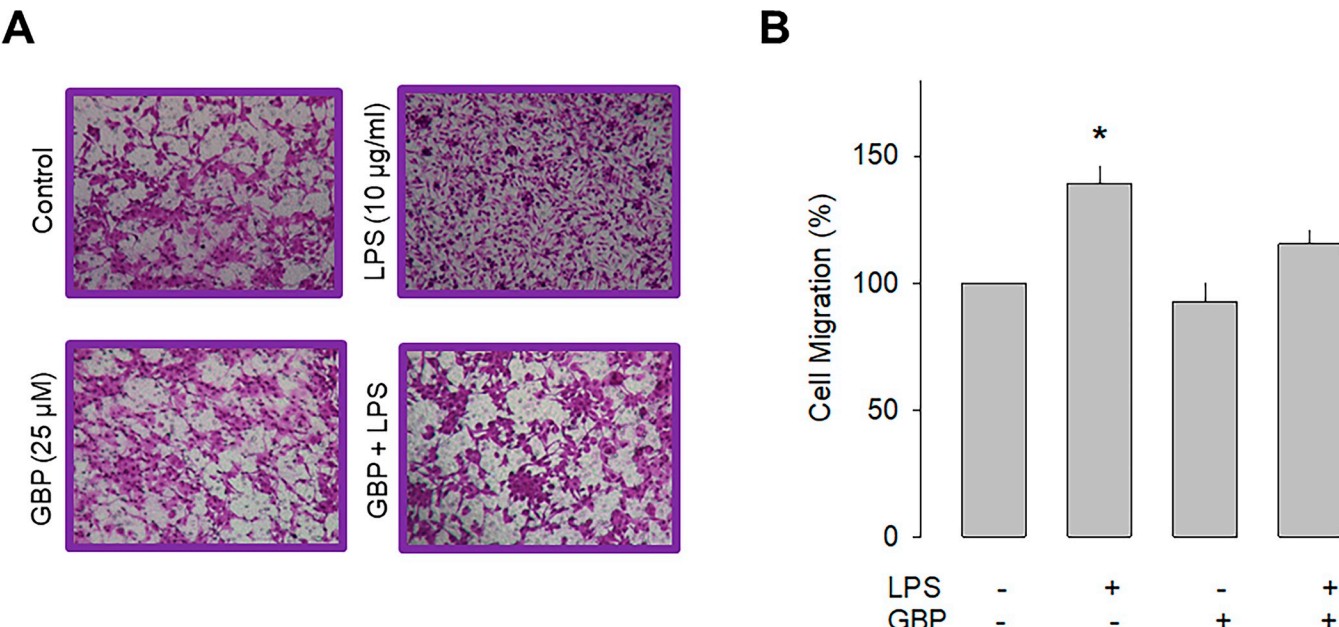

**Fig 4. Effect of gabapentin on U87 cell migration.** A) Representative images from three separate experiments of transwell migration assays in the presence or absence of the TLR-4 activator (LPS) and the antagonist of the $\alpha_2\delta$-1 subunit (gabapentin). B) Comparative analysis of the mean ± SEM of 3 transwell independent experiments. *P < 0.05 compared to untreated cells.

stimulatory effect of LPS on the migratory capacity of the neuroblastoma U87 cells and corroborated the contribution of $\alpha_2\delta$-1 to the cell migration process (Fig 4).

## Thrombospondin/$\alpha_2\delta$-1 interaction increase cell proliferation and migration

The experimental evidence suggests a direct action of LPS stimulation on $\alpha_2\delta$-1-mediated cell proliferation and migration. To further narrow down the specificity of $\alpha_2\delta$-1 involvement in these cellular events, we performed a series of experiments in the presence of thrombospondin-1, an endogenous ligand of the $\alpha_2\delta$-1 subunit [32]. Since thrombospondins are extracellular matrix molecules relevant in synaptogenesis and other cellular organization events [33], such experiments could hint at possible specific interactions between gliomablastoma cells and their cellular environment. Thus, we then determined the ability of thrombospondin to induce proliferation in U87 cells. After 48 h of stimulation, the dose-response curve presented in Fig 5A shows that TSP induces a significant increase in the proliferation of U87 cells, with half of the maximum attained at ~0.5 nM and a maximum effect observed at 1 nM. Stimulation by TSP at 1 nM increased cell proliferation at a similar level to that induced by LPS (5 μg/ml). In subsequent experiments, the effect of TSP on the migration of U87 cells was examined (Fig 5B). Hence, this analysis shows that incubation with TPS also significantly affects cell migration, producing an increase of ~25% respect to the control condition (Fig 5C). These data support the idea that the $\alpha_2\delta$-1 subunit regulates proliferation and migration in U87 cells.

## Sp1 and TLR-4 overexpression increases cell proliferation and migration

Previous studies in human tumors have shown that Sp1 and TLR-4 participate in glioblastoma development; therefore, the overexpression of these proteins was subsequently evaluated separately in U87 glioblastoma cells to assess their role in cell proliferation and migration. The cell

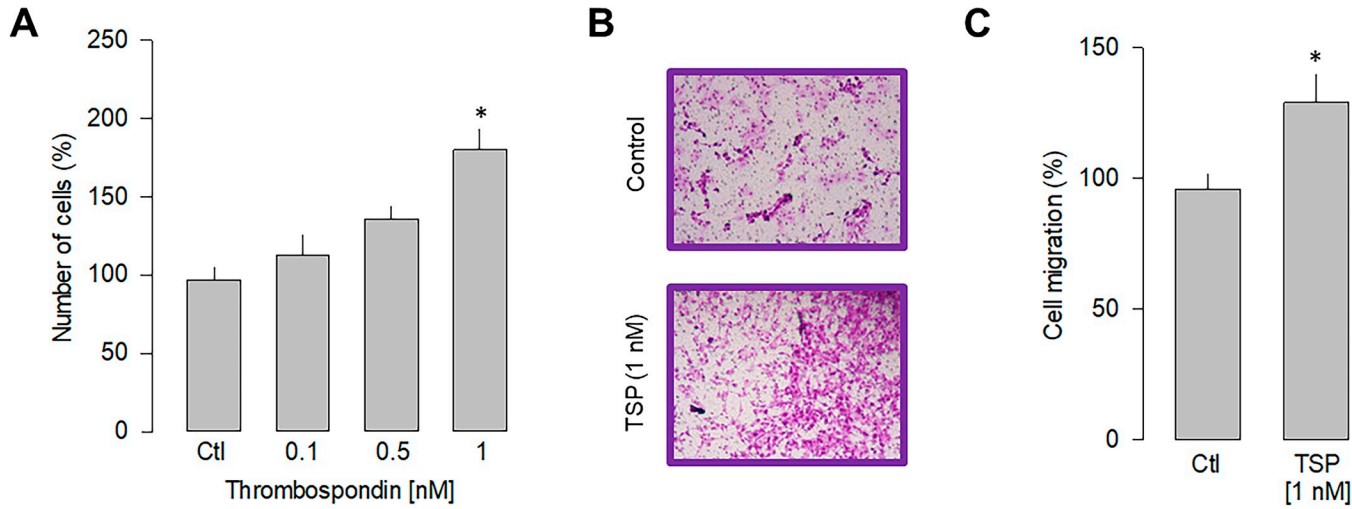

**Fig 5. Effect of thrombospondin-1 on proliferation and migration.** A) Comparison of the proliferation of U87 cells in response to various doses of thrombospondin-1 as indicated. B) Representative images from three separate experiments of transwell migration assays in the presence or absence of the thrombospondin-1 (TSP; 1 nM), an activator of the $\alpha_2\delta$-1 subunit, after 48 h of incubation. C) Comparative analysis of three transwell independent experiments as in B. Data show the mean ± SEM of three separate experiments. The asterisk denotes statistically significant differences with respect to the control condition ($^*P < 0.05$ compared to untreated cells).

counting results showed that Sp1 and TLR-4 overexpression significantly increased cell proliferation, although to a lesser extent than observed after $\alpha_2\delta$-1 overexpression (Fig 6A). Similarly, the overexpression of Sp1 and TLR-4 promoted cell migration, as observed in transwell chamber assays (Fig 6B). In this case, the effect is more evident than that produced by the overexpression of $\alpha_2\delta$-1 (Fig 6C). These findings corroborate that $\alpha_2\delta$-1 is directly involved in U87 cell proliferation and migration and suggest that the Sp1 and TLR-4 also play decisive roles in the proliferation and migration of glioblastoma cells.

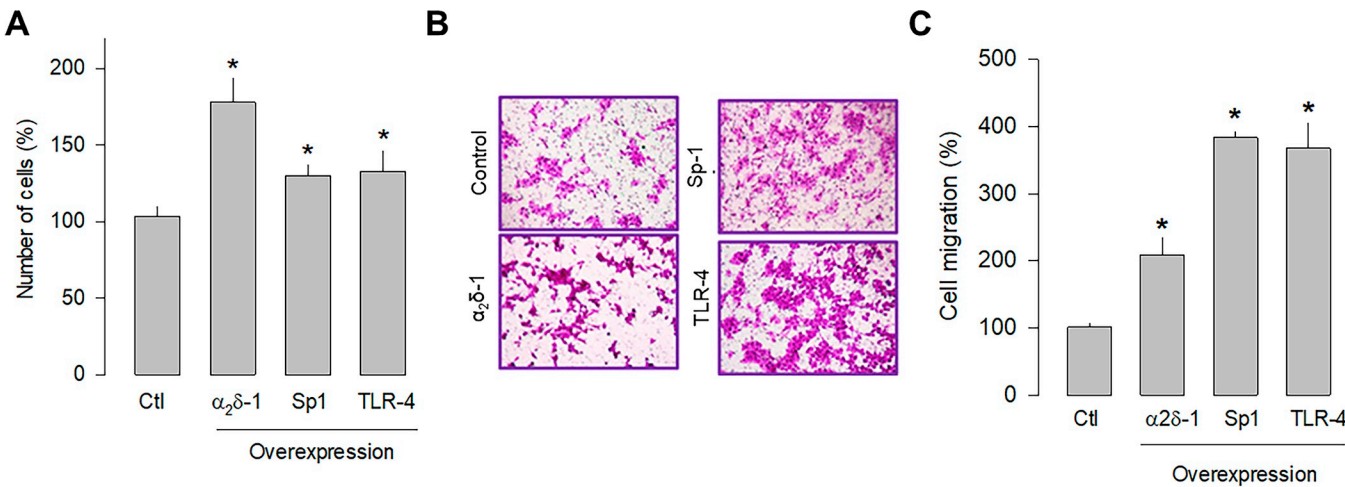

**Fig 6. Effect of $\alpha_2\delta$-1, Sp1, or TLR-4 overexpression on the proliferation and migration of U87 cells.** A) Comparison of cell number after transient transfection of U87 cells with $\alpha_2\delta$-1 subunit, Sp1, and TRL-4 as assessed by direct cell counting. Values are expressed as the percentage of untransfected cells (Ctl), and each bar represents the mean ± SEM of triplicate determinations in 3 separate experiments. $^*P < 0.05$ *versus* untransfected cells. B) Transwell migration assays of control and cells transiently transfected with $\alpha_2\delta$-1, Sp1, or TLR-4 cDNA clones. Representative images from three independent experiments are shown. C) Comparative analysis of the mean ± SEM of 3 transwell separate experiments as in B. $^*P < 0.01$ compared to untransfected cells.

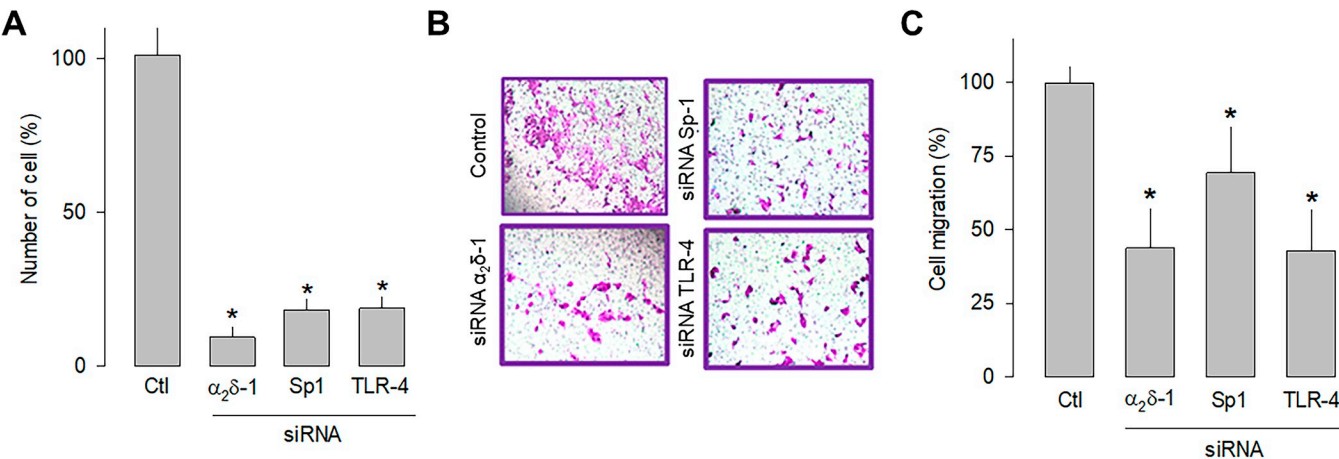

**Fig 7. Effect of α₂δ-1, Sp1, or TLR-4 silencing on the proliferation and migration of U87 cells.** A) Comparison of cell number after α₂δ-1, Sp1, and TRL-4 knockdown with specific siRNAs of U87 cells as assessed by direct cell counting. Values are expressed as the percentage of untransfected cells (Ctl), and each bar represents the mean ± SEM of triplicate determinations in 3 separate experiments. *P < 0.01 *versus* untransfected cells. B) Transwell migration assays of control and cells transiently transfected with specific α₂δ-1, Sp1, or TLR-4 siRNAs. Representative images from three separate experiments are shown. C) Comparative analysis of the mean ± SEM of 3 transwell independent experiments as in B. *P < 0.05 compared to untransfected cells.

In order to independently verify whether α₂δ-1, Sp1, and TLR-4 are related to cell proliferation and migration, a knockdown strategy for each of these proteins was then implemented. Hence, small interfering RNAs (siRNAs), were transfected into U87 cells before being subjected to proliferation (cell count) and migration (transwell chambers) assays. The results of this series of experiments showed that silencing α₂δ-1, Sp1, or TLR-4 protein expression using specific siRNAs significantly decreased the number of cells (Fig 6A) and their migration capacity (Fig 7B and 7C). These data corroborate that α₂δ-1, Sp1, and TLR-4 are related to glioblastoma cells' proliferative and migratory capacity.

## TLR-4 activation regulates α₂δ-1 expression through the NF-kB/Sp1 signaling pathway

Last, we sought to determine which signaling pathway triggered by the activation of TLR-4 could be increasing the expression of Sp1 that finally resulted in the increased expression of α₂δ-1. To this end, we first performed knockdown and overexpression experiments of TLR-4 and Sp1 separately using specific siRNAs and employing a siRNA scramble as a negative control and a siRNA against α₂δ-1 as a positive control. Then, the effect of the experimental maneuver was evaluated by estimating protein expression by Western blot. This analysis showed that the silencing of both TLR-4 and Sp1 significantly decreased the expression of α₂δ-1 (Fig 8A). Conversely, by inducing the overexpression of both TLR-4 and Sp1, the expression of α₂δ-1 was significantly increased (Fig 8B).

The second series of experiments were carried out using an inhibitor of the nuclear factor κB (NF-κB) since it is known that the activation of TLR-4 involves two signaling pathways, a canonical one dependent on MyD88 (myeloid differentiation factor 88), and another non-canonical dependent on a protein called TRIF (Toll/IL-1R domain-containing adapter-inducing IFN-β). However, both pathways converge the nuclear translocation of NF-kB, affecting different transcription factors, including Sp1. For this reason, the expression of α₂δ-1 and Sp1 was then evaluated in the presence and absence of the NF-κB inhibitor PDTC (pyrrolidine dithiocarbamate; Fig 9). These experiments showed that the inhibition of NF-kB significantly decreased the expression of both proteins. Moreover, this effect was maintained even in the

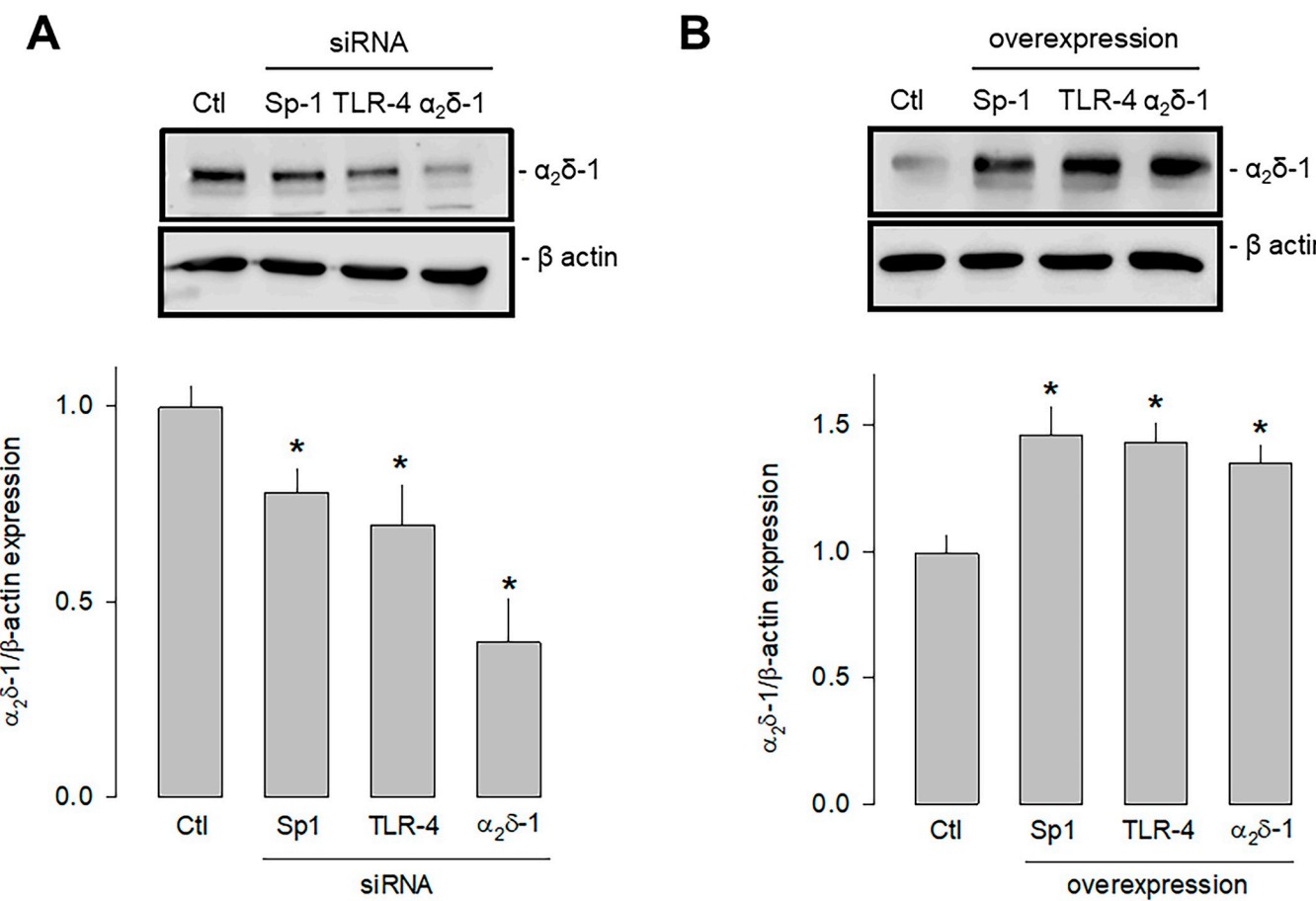

**Fig 8. Effect of the knockdown and overexpression of TLR-4 and Sp1 on α₂δ-1 expression.** A) Western blot assays performed on U87 cell lysates in the control condition and after transfection with the Sp1, TLR-4, and α₂δ-1 specific siRNAs. The image shows a representative assay of three performed separately (upper panel). The signal obtained with the β-actin antibody served as the loading control. The lower panel shows the comparative densitometric analysis of the level of α₂δ-1 protein. Data are shown as mean + SEM of three independent experiments in triplicate. B) Western blot assays performed on U87 cells in the control condition and after transfection with the Sp1, TLR-4, and α₂δ-1 cDNA clones. The image shows a representative assay of three performed separately (upper panel). The signal obtained with the β-actin antibody served as the loading control. The lower panel shows the comparative densitometric analysis of the level of α₂δ-1 protein. Data are shown as mean + SEM of 3 independent experiments in triplicate. *P < 0.05 compared to untransfected cells.

presence of LPS, which supports the idea that the activation of TLR-4, which triggers the formation of the IKK/NF-κB complex, regulates the expression of Sp1 and in consequence the expression of α₂δ-1.

## Discussion

The α₂δ-1 subunit is a protein with complex topological features, having a molecular weight of ~170 kDa subject to post-translational proteolytic cleavage. Hence, the mature protein is composed of two peptides, α₂ and δ, which in native conditions are kept together by a disulfide bond [34]. Although the central physiological role of α₂δ-1 is to function as an auxiliary subunit of voltage-gated $Ca^{2+}$ channels [30], its significant role in tumor progression has already been demonstrated previously in several types of cancer cell lines, including larynx, lung, ovary, liver, stomach, and breast [13–20]. However, its contribution to the onset and/or development of nervous system tumors remains virtually unexplored. In this report, we show evidence that TLR-4 activation by LPS increases the expression of the α₂δ-1 subunit and promotes the proliferative and migratory potential of U87 human glioblastoma cells.

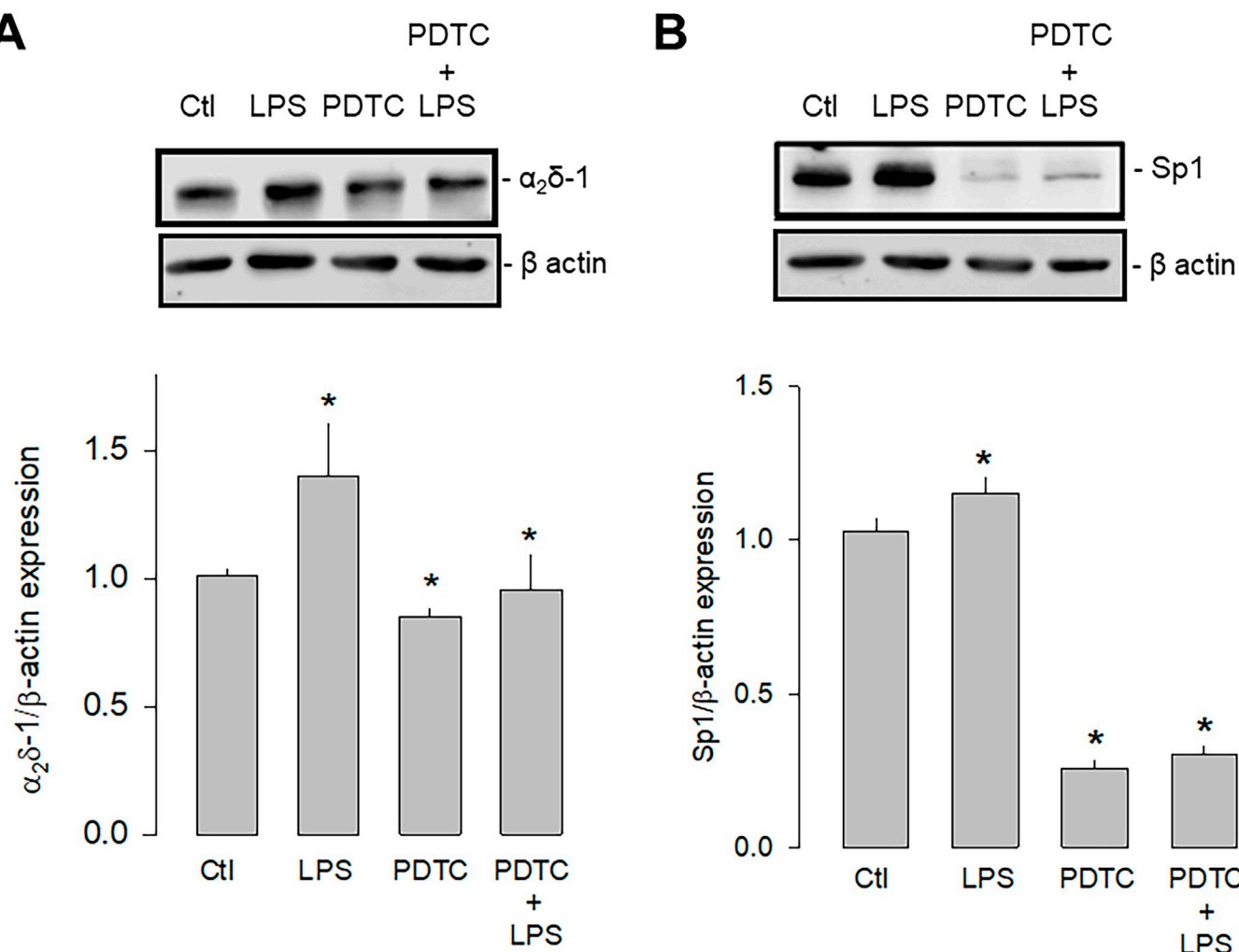

**Fig 9. Effect of NF-κB inhibition on Sp1 and α₂δ-1 expression.** A) Western blot assays performed on U87 cell lysates in the control condition and after treatment with the TLR-4 agonist LPS and the transcription factor NF-κB inhibitor PDTC. The image shows a representative assay of three performed separately (upper panel). The signal obtained with the β-actin antibody served as the loading control. The lower panel shows the comparative densitometric analysis of the level of α₂δ-1 protein. Data are shown as mean + SEM of 3 independent experiments in triplicate. B) Sp1 Western blot assays performed on U87 cells in the conditions as in A. The image shows a representative assay of three performed separately (upper panel). The signal obtained with the β-actin antibody served as the loading control. The lower panel shows the comparative densitometric analysis of the level of Sp1 protein. Data are shown as mean ± SEM of 3 independent experiments in triplicate. $^*P < 0.05$ compared to untreated cells.

Furthermore, our data suggest that TLR-4-mediated regulation of α₂δ-1 expression occurs through the NF-kB/Sp1 signaling pathway.

Toll-like receptors (TLRs) signaling pathways play essential roles in the immune system controlling tumor growth and progression. TLRs activation may result in the production of cytokines, chemokines, interferons, and the transcription factor NF-κB. TLR-4 activation may be mediated by LPS, a bacterial endotoxin, leading to two distinct signaling pathways: the primary response pathway through MyD88 and the IFN-β (TRIF) pathway [35, 36]. Activation of TLR-4 may increase the production of factors that promote tumor development via the MyD88 pathway. MyD88 activates the IRAK kinase and interacts with the TRAF6 factor, resulting in the nuclear translocation of NF-κB, which, once in the nucleus, can interact and coactivate the Sp1 transcription factor promoting the transcription of genes, including that of the α₂δ-1subunit.

It is worth mentioning here that voltage-gated ion channels in general, and $Ca^{2+}$ channels in particular, play fundamental roles in controlling electrical signaling through the generation of nerve impulses [37]. In addition, however, it has been shown that these proteins can also contribute significantly to cellular biochemical signaling, helping to determine several physiological events, including cell division, cell cycle progression, and volume regulation [38]. All these functions are of great relevance for the proliferation of cancer cells [37–39].

The intracellular $Ca^{2+}$ concentration is precisely and strictly controlled to generate $Ca^{2+}$ signals with particular spatio-temporal attributes. Furthermore, this control is essential for the differential regulation of different proteins and $Ca^{2+}$-dependent signaling pathways involved in specific cellular processes, including cell proliferation and migration [7, 39]. Since these functions are relevant for tumorigenesis, alterations in intracellular $Ca^{2+}$ homeostasis and $Ca^{2+}$ signaling have been proposed to be crucial in driving or maintaining the malignant phenotypes. Indeed, tumor transformation is associated with changes in the expression and/or function of the molecules responsible for $Ca^{2+}$ transport, which eventually controls many intracellular signaling pathways. This may result in the evasion of apoptosis with increased survival, excessive proliferation, malignant angiogenesis, cell migration, and metastasis [7]. In this context, the overexpression of $\alpha_2\delta$-1 could be causing changes in membrane localization of $Ca_V$ channels in U87 cells, which could give rise to changes in the $Ca^{2+}$ concentration of different compartments and cellular microdomains and changes in intracellular signaling patterns.

On the other hand, it has been proposed that other $\alpha_2\delta$ subunits (of which there are four isoforms) are also associated with cancer development. For example, recent studies in prostate cancer have shown that $\alpha_2\delta$-2 overexpression induces increased cell proliferation *in vitro* and that the overexpression of $\alpha_2\delta$-2 LNCaP cell xenografts in nude mice is tumorigenic. In support of a direct role for $\alpha_2\delta$-2 in prostate cancer development, gabapentin, an inhibitor of $\alpha_2\delta$ subunits, was capable of inhibiting tumor development in xenografts [40].** In contrast, the *CACNA2D3* gene, which encodes for $\alpha_2\delta$-3, has been identified as a methylation site in gastric cancer. Furthermore, *CACNA2D3* methylation is associated with shorter survival, making it a useful prognostic marker for this type of cancer [41]. The potential tumor-inhibiting properties of $\alpha_2\delta$-3 have also been observed *in vitro*, where overexpression results in reduced cell growth and adhesion, while its silencing using siRNAs had the opposite effect [41]. Likewise, the methylation-dependent silencing of *CACNA2D3* has been proposed as a biomarker for the risk of metastasis in breast cancer [42]. These findings imply that, beyond the canonical role of the $\alpha_2\delta$ auxiliary subunits in determining the properties and functional diversity of the high-threshold voltage-gated $Ca^{2+}$ channels, there may be some non-canonical molecular interactions not related to its primary function as part of the macromolecular complex of $Ca^{2+}$ channels. These interactions could be responsible, at least in part, for some of the actions of the $\alpha_2\delta$-1 subunit on cell proliferation and migration.

Though the precise molecular mechanism by which the $\alpha_2\delta$-1 subunit affects proliferation and migration in human glioblastoma cells, its cellular actions may be grouped into two general mechanisms. The first of them would be directly associated with the function of the subunit as part of the high-threshold $Ca^{2+}$ channel complex, while the second would be independent of its role in the channels. The elucidation of which of these mechanisms, which are not mutually exclusive, contributes to a greater extent to the actions of $\alpha_2\delta$-1 on cell proliferation and migration in glioblastoma cells, as well as the identification of molecular interactors for the $\alpha_2\delta$-1 subunit outside the channel related to the control of cell proliferation and migration, are undoubtedly exciting topics for future research. In any case, the link found between the $Ca^{2+}$ channel auxiliary subunit and glioblastoma reveals this protein as a possible biomarker and/or therapeutic target. However, further mechanistic understanding of how the

$\alpha_2\delta$-1 contributes to tumor progression will be required before its diagnostic or therapeutic potential can be exploited.

## Supporting information

**S1 Fig. Expression of α2δ-1, TLR-4, and Sp1 in human glioblastoma U87 and human neuroblastoma SHSY-5Y (SH) cell lines.** The image shows representative blots of three experiments performed separately. The signal obtained with the β-actin antibody served as the loading control.
(TIF)

**S1 Raw images.**
(PDF)

## Acknowledgments

The authors are grateful to the support and assistance provided by Mercedes Urbán.

## Author Contributions

**Conceptualization:** Miriam Fernández-Gallardo, Ricardo González-Ramírez, Rodolfo Delgado-Lezama, Eduardo Monjaraz, Ricardo Felix.

**Data curation:** Ricardo González-Ramírez, Ricardo Felix.

**Formal analysis:** Miriam Fernández-Gallardo, Ricardo Felix.

**Investigation:** Miriam Fernández-Gallardo, Alejandra Corzo-Lopez, David Muñoz-Herrera, Margarita Leyva-Leyva, Ricardo González-Ramírez, Alejandro Sandoval, Eduardo Monjaraz.

**Methodology:** Miriam Fernández-Gallardo, Alejandra Corzo-Lopez, David Muñoz-Herrera, Margarita Leyva-Leyva, Ricardo González-Ramírez, Alejandro Sandoval, Eduardo Monjaraz.

**Project administration:** Ricardo Felix.

**Supervision:** Rodolfo Delgado-Lezama, Eduardo Monjaraz, Ricardo Felix.

**Writing – original draft:** Miriam Fernández-Gallardo, Ricardo Felix.

**Writing – review & editing:** Ricardo González-Ramírez, Alejandro Sandoval, Rodolfo Delgado-Lezama, Eduardo Monjaraz, Ricardo Felix.

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
