## [Decision Letter · Decision Letter 0]

19 Jul 2022

PONE-D-22-09312Role of the Ca2+ channel α2δ-1 auxiliary subunit in proliferation and migration of human glioblastoma cellsPLOS ONE

Dear Dr. Felix,

Thank you for submitting your manuscript to PLOS ONE. After careful consideration, we feel that it has merit but does not fully meet PLOS ONE’s publication criteria as it currently stands. Therefore, we invite you to submit a revised version of the manuscript that addresses the points raised by the reviewer. While revising your manuscript, please provide the dot plots instead of all bar graphs and submit uncropped Western blots images in Supplemental materials.

We look forward to receiving your revised manuscript.

Kind regards,

Alexander G Obukhov, Ph.D.

Academic Editor

PLOS ONE

Journal Requirements:

Unfunded study

NO authors have competing interests

Reviewers' comments:

Reviewer's Responses to Questions

**Comments to the Author**

1. Is the manuscript technically sound, and do the data support the conclusions?

Reviewer #1: Yes

2. Has the statistical analysis been performed appropriately and rigorously? 

Reviewer #1: Yes

3. Have the authors made all data underlying the findings in their manuscript fully available?

Reviewer #1: Yes

4. Is the manuscript presented in an intelligible fashion and written in standard English?

Reviewer #1: Yes

5. Review Comments to the Author

Reviewer #1: The study by Fernández-Gallardo et al. explore the expression of α2δ1-protein and related proteins in glioblastoma cells to elucidate the impact of α2δ1 in cell proliferation. Based on indications in the literature the authors explored to which extend α2δ1 expression correlates with the proliferation of U87 cells. Here they used pharmacological antagonists and siRNA mediated modulation of α2δ1 expression to explore the impact of the α2δ1 protein on cell proliferation and intracellular Nfkb mediated signalling cascades. The use of the glioma cell line provides a good model for a channel independent exploration of the α2δ1 mediated aspects in cell proliferation/migration apart from the classical function as calcium channel accessory subunit.

The data suggest a direct action of α2δ1 protein expression as response to LPS stimulation. To further narrow the specificity of LPS induced proliferation a positive control via the application of Thrombospondin-1 would strengthen the correlation between the stimulus and the impact of α2δ1-proteins as receptor molecule of Throbospondin-1. Since Thrombospondins are also for neurons relevant ECM molecules that influence synaptogenesis, such experiment could hint to specific possible interactions between glioma cells and their cellular environment.

Within the western blot analysis of expression, the data have been normalized to the control, but the controls missing an indication of the standard error. To better interpret the significance of the effects such information would be very helpful to see how reliable/stable are the western blot experiments. Please include such error information into the graphs.

6. PLOS authors have the option to publish the peer review history of their article (what does this mean?). If published, this will include your full peer review and any attached files.

Reviewer #1: No

---

## [Author Response · Author response to Decision Letter 0]

18 Nov 2022

Response to Reviwer’s comments:

Reviewer #1: 

Comment 1. 

The data suggest a direct action of α2δ1 protein expression as response to LPS stimulation. To further narrow the specificity of LPS induced proliferation a positive control via the application of Thrombospondin-1 would strengthen the correlation between the stimulus and the impact of α2δ1-proteins as receptor molecule of Throbospondin-1. Since Thrombospondins are also for neurons relevant ECM molecules that influence synaptogenesis, such experiment could hint to specific possible interactions between glioma cells and their cellular environment.

Response to Comment 1. 

As the reviewer points out, the use of Thrombospondin-1, a ligand of the a2d-1 subunit, would help to strengthen the proposal that this protein could significantly contribute to determining the proliferation and migration processes in the cell line derived from glioblastoma. In fact, following the reviewer's suggestion, we have performed additional experiments to evaluate the effects of Thrombospondin-1 on the proliferation and migration of U87 cells. Our results suggest that incubation with Thrombospondin-1 for 24 h significantly increases both cellular processes. This analysis has been included as a new figure (see Fig 5) in the revised version of the manuscript. Accordingly, a detailed description of this new series of experiments has been included in both the Results section (p. 12; l. 246-255) sections. We thank the reviewer for the suggestion that has undoubtedly strengthened our interpretation that a2d-1 participates in cell proliferation and migration processes.

Comment 2. 

Within the western blot analysis of expression, the data have been normalized to the control, but the controls missing an indication of the standard error. To better interpret the significance of the effects such information would be very helpful to see how reliable/stable are the western blot experiments. Please include such error information into the graphs.

Response to Comment 2. 

Corrected. In order to provide more information on the interpretation of the results, following the reviewer's suggestion in the manuscript version, the error bars for the control condition have been included in all bar graphs included in the paper (see Figs 1A and 1C; Figs 2A, 2B, 2D; Fig 3; Fig 4B; Figs 5A and 5C; Figs 6A and 6C; Figs 7A and 7C; Figs 8A and 8B; Figs 9A and 9B). Once again, we thank the reviewer for the suggestion.

---

## [Decision Letter · Decision Letter 1]

2 Dec 2022

Role of the Ca2+ channel α2δ-1 auxiliary subunit in proliferation and migration of human glioblastoma cells

PONE-D-22-09312R1

Dear Dr. Felix,

We’re pleased to inform you that your manuscript has been judged scientifically suitable for publication and will be formally accepted for publication once it meets all outstanding technical requirements.

Kind regards,

Alexander G. Obukhov, Ph.D.

Academic Editor

PLOS ONE

Reviewers' comments:

Reviewer's Responses to Questions

**Comments to the Author**

1. If the authors have adequately addressed your comments raised in a previous round of review and you feel that this manuscript is now acceptable for publication, you may indicate that here to bypass the “Comments to the Author” section, enter your conflict of interest statement in the “Confidential to Editor” section, and submit your "Accept" recommendation.

Reviewer #1: (No Response)

2. Is the manuscript technically sound, and do the data support the conclusions?

Reviewer #1: Yes

3. Has the statistical analysis been performed appropriately and rigorously? 

Reviewer #1: Yes

4. Have the authors made all data underlying the findings in their manuscript fully available?

Reviewer #1: Yes

5. Is the manuscript presented in an intelligible fashion and written in standard English?

Reviewer #1: Yes

6. Review Comments to the Author

Reviewer #1: The paper focus on the function of a2d1-subunit as a proliferation stimuli within U87 human gliablastoma cells. Following several hypothetical molecular pathways the authors show a connection between the toll-like receptor TLR-4, the transcription factor Sp1 and a2d1, as one of the target genes for this transcription factor. The proliferation effect was measured and modulated either by overexpression or downregulation of interacting proteins or pharmacological treatments. Here gabapentin and thrombospondin were used to block or stimulate a2d1-subunits, respectively. The essays used are robust and all experiments have been done with the needed controls.

The study show additional channel independent function of a2d-proteins, as seen also in neurons and impose the question about the mechanism of this channel independent signalling, which hopefully is subject of further studies.

7. PLOS authors have the option to publish the peer review history of their article (what does this mean?). If published, this will include your full peer review and any attached files.

Reviewer #1: No

---

## [Editor Report · Acceptance letter]

6 Dec 2022

PONE-D-22-09312R1 

Role of the Ca2+ channel α2δ-1 auxiliary subunit in proliferation and migration of human glioblastoma cells 

Dear Dr. Felix:

I'm pleased to inform you that your manuscript has been deemed suitable for publication in PLOS ONE. Congratulations! Your manuscript is now with our production department. 

Kind regards, 

on behalf of

Dr. Alexander G Obukhov 

Academic Editor

PLOS ONE